# An Approach to Traumatic Brain Injury-Related Hypopituitarism: Overcoming the Pediatric Challenges

**DOI:** 10.3390/diagnostics13020212

**Published:** 2023-01-06

**Authors:** Raluca Maria Vlad, Alice Ioana Albu, Irina Delia Nicolaescu, Ruxandra Dobritoiu, Mara Carsote, Florica Sandru, Dragos Albu, Daniela Păcurar

**Affiliations:** 1Department of Pediatrics, “Grigore Alexandrescu” Emergency Children’s Hospital, 011743 Bucharest, Romania; 2Faculty of Medicine, “Carol Davila” University of Medicine and Pharmacy, 020021 Bucharest, Romania; 3Department of Endocrinology, “Carol Davila” University of Medicine and Pharmacy, 020021 Bucharest, Romania; 4“Elias” Emergency Clinical Hospital, 011461 Bucharest, Romania; 5St. Mary Medical Clinic, 011172 Bucharest, Romania; 6“C. I. Parhon” National Institute of Endocrinology, 011863 Bucharest, Romania; 7Department of Dermatovenerology, “Carol Davila” University of Medicine and Pharmacy, 020021 Bucharest, Romania; 8“Elias” University Emergency Hospital, 011461 Bucharest, Romania; 92nd Clinical Department Obstetrics Gynecology, Faculty of Dentistry, “Carol Davila” University of Medicine and Pharmacy, 020021 Bucharest, Romania

**Keywords:** hypopituitarism, traumatic brain injury, growth hormone deficiency, delayed puberty, ACTH, height, TSH

## Abstract

Traumatic brain injury (TBI)-related hypopituitarism is a rare polymorphic complication of brain injury, with very little data, particularly concerning children and teenagers. This is a comprehensive review of the literature regarding this pathology, starting from a new pediatric case. The research was conducted on PubMed and included publications from the last 22 years. We identified nine original studies on the pediatric population (two case reports and seven studies; only four of these seven were prospective studies). TBI-related hypopituitarism is associated with isolated hormonal deficits ranging from 22.5% to 86% and multiple hormonal deficiencies from 5.9% to 50% in the studied pediatric population. Growth hormone (GH) deficiency is most often found, including the form with late occurrence after TBI; it was described as persistent in half of the studies. Thyroid-stimulating hormone (TSH) deficiency is identified as a distant complication following TBI; in all three studies, we identified this complication was found to be permanent. Adrenocorticotropic hormone (ACTH) deficiency did not relate to a certain type of brain trauma, and it was transient in reported cases. Hyperprolactinemia was the most frequent hormonal finding, also occurring late after injury. Central diabetes insipidus was encountered early post-TBI, typically with a transient pattern and did not relate to a particular type of injury. TBI-related hypopituitarism, although rare in children, should be taken into consideration even after a long time since the trauma. A multidisciplinary approach is needed if the patient is to safely overcome any acute condition.

## 1. Introduction

Traumatic brain injury (TBI) is a common condition, with 69 million people suffering from this pathology each year worldwide [1]. TBI has potentially serious consequences, being a major cause of mortality and morbidity in both the developed and developing world [1]. TBI can be of variate degrees of severity according to Glasgow Coma Scale, being classified as mild, moderate and severe [2]. Although the probability of long-term sequelae is higher in patients with severe TBI, those with mild and moderate TBI may experience post-concussion syndrome with chronic headaches, cognitive deficits, and memory and behavioral issues [3]. However, some of these symptoms might be the consequence of hypopituitarism, another possible complication of TBI, but with a particular diagnosis and therapeutic approach [4]. 

There is a wide range of reported prevalence of TBI-related hypopituitarism (5–61%) [5,6]. The lack of clear data is probably a result of the varied design of the studies and the heterogeneity of the studied populations [7,8,9]. However, the exact prevalence is difficult to be known since most of the studies focused on patients with severe and moderate TBI who were admitted to the hospital, the occurrence of hypopituitarism in those with mild TBI being largely unknown [10]. Moreover, the lack of specific guidelines for hypopituitarism assessments in patients with previous TBI, also concerning long-term follow-up and serial check-ups, may result in missed or late diagnosis with potentially severe consequences [11,12,13]. 

Regarding the development of TBI-related hypopituitarism, the exact timing of hormonal deficiencies emerging after acute brain trauma is less understood. It can occur early after a TBI, but there are studies that mention a very long time gap between acute injury and the onset of symptoms (5 months up to 15 years) [14]. However, signs of hormone deficiencies may appear during the first year after a TBI. In a study published by Bondanelli et al., about 40% of patients with a history of TBI developed, in most cases, isolated pituitary deficiencies and rarely complete pituitary failure. Altered secretion of somatotropin and gonadotropin seemed to be the most common problem, followed by deficiency of Adrenocorticotropic Hormone (ACTH) and Thyroid Stimulating Hormone (TSH) [15].

Regarding the causal relationship between the severity of brain trauma and the degree of TBI-related hypopituitarism, we do not have enough data so far. Some would suggest that mild TBI is associated with the low occurrence of hypopituitarism, but inter-observer and time variability of the Glasgow Coma Scale Score suggests that this measurement does not predict the risk of TBI-related hypopituitarism [16]. 

The pituitary involvement is the result of hemorrhage, infarction or shearing injury lesions, followed by hypoxic insults and an inflammatory reaction [17,18,19,20,21]. Both anterior and posterior pituitary can be affected, with the gonadotropes and Growth Hormone (GH) being the most commonly involved [9,22]. Isolated or multiple pituitary hormone deficiencies are either transient or persistent, complicating the development of screening recommendations [23,24,25]. Their presence can impact normal growth and pubertal development in children and teenagers, associating significant consequences on the quality of life, but with a potentially lethal impact due to unrecognized and untreated ACTH deficiency. Thus, identifying TBI-related hypopituitarism in children might be of paramount importance for normal development and overall health [26].

The aim of this paper is to highlight the importance of early identification of pituitary abnormalities in young patients through a case report and to review the current literature regarding hormone deficits diagnosed secondary to brain trauma in children and adolescents. 

## 2. Methods

This literature review is based on PubMed papers published in the last 22 years (from 2000 to 2022) regarding hormonal changes among patients who experienced brain trauma, with a primary interest in the pediatric field. Particular search terms (different combinations), such as “posttraumatic hypopituitarism”, “hormonal deficiencies”, “pituitary function after brain injury”, “traumatic brain injury related hypopituitarism”, or “hormonal dysfunction in children after brain trauma”, were used in order to find the most suitable studies for this review. The inclusion criteria were: a definite diagnosis of TBI-related hypopituitarism, confirmed hormonal dysfunction of at least one pituitary line, and a follow-up period of at least 1 year post-TBI. The exclusion criteria were: incomplete patient history or diagnostic workup, non-English published papers, and adults with TBI.

The authors also present a complex illustrative case report managed by a multidisciplinary team. The diagnosis of hypopituitarism was established a long time after TBI. Written informed consent to publish this paper is obtained from the patient and his parents.

## 3. Case Report

Hereby, we present the case of a teenager with a late diagnosis of TBI-related hypopituitarism and the evolution under hormonal treatment in order to underscore the importance of timely diagnosis. 

A 17-year-old male patient was admitted to the Orthopedics Department for surgical correction of a chronic right hip epiphysiolysis due to a car accident at the age of 13 years when he also had a severe brain injury with multiple skull base fractures and respiratory support for 1.5 months. The laboratory tests showed liver cytolysis-alanine transaminase and aspartate transaminase (three times the upper limit of the normal range), so he was first transferred to the Department of Pediatrics to further investigations and management before surgery. 

The physical examination revealed proportionate short stature with a height of 157 cm and a standard deviation (SD) score for a height of -3.1 SD, according to Romanian references [27]. The patient had a normal body mass index (BMI) of 24.5 kg/m^2^ 0.96 SD according to WHO (World Health Organization) standards [28]. He presented fatigability, loss of right eyesight, coarse features, dry, harsh skin, coarse scalp hair, no axillary and facial hair, pubertal stage Tanner P1G2 (testicular volume of 6 mL), blood pressure of 90/60 mmHg, heart rate of 60/minute, normal thyroid gland at physical examination, and inferior limb inequality. Visual field and visual acuity were normal in the left eye (Figure 1).

Based on the clinical picture, the suspicion of hypogonadism was raised, and the evaluation of the patient was continued in a multidisciplinary team (pediatrician and endocrinologist). The hormone evaluation revealed central hypothyroidism, adrenal insufficiency, low testosterone and low gonadotrophins: luteinizing hormone (LH), respective follicle-stimulating hormone (FSH), and reduced insulin-like growth hormone 1 (IGF-1) (Table 1).

Imaging findings included thyroid ultrasound describing a thyroid gland of 1.3 by 1.1 by 3.2 cm (right lobe), 0.8 by 1.2 by 2.6 cm (left lobe), and 1.5 by 0.3 cm (isthmus) with a homogeneous structure (a relatively small gland) [29]. X-ray of the non-dominant hand showed a delayed bone age of 13.5 years.

Based on the clinical and lab data, the following diagnosis was established: secondary hypothyroidism, hypogonadism and adrenal insufficiency and short stature, most probably as elements of post-TBI hypopituitarism. Central nervous system magnetic resonance imaging (MRI) with contrast media confirmed post-traumatic aspects: a small pituitary gland (~1.2 mm), right deviated pituitary stalk with neurohypophysis in addition to other post-traumatic brain lesions like discontinued right optic nerve, para-sphenoidal herniation of the arachnoid (Figure 2).

The ophthalmologic evaluation described post-traumatic atrophy of the right optic nerve with right eye vision loss. Substitution treatment was initiated with Prednisone 5 mg daily and levothyroxine 50 µg daily. Laboratory assays showed high but decreasing values of transaminases during the days following admission. Other causes of liver cytolysis were excluded (for instance, chronic viral hepatitis, Wilson disease, and autoimmune hepatitis). After the correction of glucocorticoid and thyroid hormone deficiencies, the GH reserve was tested with a clonidine test. GH values during the clonidine-stimulating test were found to be below the value of 0.05 ng/mL (with a normal value of 7 ng/mL referring to a maximal GH peak at any time). Thus, the diagnosis of GH deficiency was confirmed, and somatropin treatment was started at a dose of 0.03 µg/kg body weight/day, followed by initiation of transdermal testosterone 23 mg/day. During the follow-up period of more than 3 years, the patient gained 2.55 SD in height with a bone age of 14 years at the age of 20.5 years. His actual height is 173.5 cm at the age of 20.5 years (−0.55 SD). The levothyroxine dose was gradually increased to 100 µg/day, and the dose of transdermal testosterone was increased to 46 mg/day after 2 years of treatment. Permanent hypopituitarism requires lifelong follow-up and period checkups, including glucocorticoid replacement adjustments during surgeries, infections, etc. 

## 4. Review of the Literature

All papers concerning TBI were reviewed (which were published since 2000). Nine of the 80 publications included data regarding children. We found two clinical cases and seven pediatric studies that followed-up patients with TBI-related hypopituitarism at least 1 year after brain injury. In these seven studies, we assessed the number of patients, gender distribution, median age, trauma type and severity, the interval between the TBI and the associated hormonal panel assessments, the hormonal deficits (isolated or multiple) and their prevalence. Only four of these seven were prospective studies. According to the current reviewed literature, post-TBI hypopituitarism occurs fairly frequently in children, with isolated hormonal deficits having a prevalence ranging from 22.5% to 86% and multiple hormonal deficiencies with a ratio between 5.9% and 50% [6,30,31,32]. 

GH deficiency was reported in six out of the even studies performed in children, with a prevalence between 4.9% and 27.8% (three studies were retrospective, and four were prospective). Data regarding the GH-IGF-1 axis in the early post-traumatic phase in children and adolescents are limited. Ulutabanca et al. [33] showed that IGF-1 levels were significantly lower in the acute phase in comparison with the chronic period. Moreover, it seems that in most cases, low IGF-1 values are transient and correlated with the severity of the trauma [33,34].

The long-term evolution of GH secretion in post-TBI status was evaluated in a few studies. Agha et al. [31] included children (and adults): from the nine patients with early GH deficiency, five had recovered within the first 6 months, but the other two patients developed new GH deficiencies. At 12 months, only five patients had persistent GH deficiencies. In the study by Auble et al. [31], 17% of the 14 children with moderate-to-severe TBI presented with low peak GH during stimulation tests [31]. The median age of the children included in the study was 3.1 years, and the median age at the injury was 5 months. Kaulfers et al. [35] prospectively studied 31 children after TBI for 1 year. They found a prevalence of 5% for GH deficiency at 1 year of follow-up, with an increasing trend from 1 month to 6 months but decreasing until 1 year. A recent study aimed to determine the prevalence of permanent pituitary hormone deficiency several years after severe TBI; the study included 61 children who were followed for at least 5 years after TBI. They found that 17 of 61 patients had GH deficiency 1 year after TBI (five patients had persistent GH insufficiency, and a new patient was diagnosed 6.5 years after TBI) [36]. Ulutabanca et al. [33] reported the presence of GH deficiency in 9.1% of the children after 1 year after TBI. Krahulic et al. [34] found that GH deficiency can be diagnosed after the first year following brain injury. This data suggest that GH deficiency may be an early endocrine dysfunction but also a late complication of TBI, occurring years after trauma or being recognized years after trauma (as seen in our case) [37]. Although some of the patients recover the GH-IGF-1 function, others can have permanent damage to the GH-producing cells without a clear identification of potential risk factors for a persistent or late onset anomaly. Some studies established a correlation between severe brain injury and the presence of GH deficiency [38,39]. 

TSH deficiency is reported with a prevalence of 3.2–42.8% [31,33,34,35,36]. In the studies conducted by Auble et al. [31], Kaulfers et al. [35], and Dassa et al. [36], TSH anomalies are identified as a late onset complication (starting with the first 6–12 months after trauma) and central hypopituitarism was found to be a permanent complication. Ulutabanca et al. [33] mentioned seven cases of TSH deficit, which occurred in the acute phase after brain injury; they were all transient and had no connection to a specific type of trauma. In the study driven by Krahulik et al. [34], out of three patients with TSH deficiency, two had suffered from severe brain damage and had multiple hormonal anomalies which persisted long after the acute phase ended and eventually deceased; the other patient developed secondary hypopituitarism 1 year after injury. In our case, the patient was stationary after the acute phase; it is difficult to identify the exact timing of hypopituitarism’s elements which were all permanent according to the more than 3-year surveillance period in addition to 4 years from TBI to the pituitary insufficiency diagnosis. 

Adrenal insufficiency with late occurrence after TBI was found to have variable ratios in pediatric studies [5,17,40,41], with a significant rate of remission according to some adult studies [42,43]. ACTH deficiency has a prevalence of 1.6–26.8% [30,33,34,36]. Agha et al. [30]. and Dassa et al. [36] mentioned it was transient and did not appear to relate to a certain type of brain trauma. Ulutabanca et al. [33] showed that ACTH deficiency occurred during the acute phase; it was transient and associated with severe brain lesions, though it was more likely a physiological response to stress/therapy rather than a real connection with a particular type of injury; in one case, the ACTH deficit developed late after brain trauma, and it was persistent. Krahulik et al. [34] described persistent low ACTH levels after the acute phase of trauma only in two patients who were deceased; for the other subjects, the deficit was transient. 

Traumatic brain injury affecting the hypophysis can lead to secondary hypogonadism [44]. In one large study including 102 patients from both the pediatric and adult populations who suffered moderate to severe brain injury, hypogonadotropic hypogonadism was the second most frequent complication that emerged. It developed in most cases 3 months after the incident, and it was usually transient. Long-term untreated hypogonadism appears to affect primarily males, and its effects rest on fertility, psychosexual function and general well-being. It also affects muscle performance and physical exercise tolerance, the level of energy and motivational status, and not least, the mortality rate secondary to cardiovascular disease [30].

Hypogonadism had a prevalence varying from 2.4 to 11.8% [30,33,34,35,36,37,38,39,40,41,42,43,44,45]. Krahulik et al. [34] discovered this hormonal anomaly late after brain trauma (12 months) in two girls who experienced irregular menstrual cycles and oligo-menorrhea. Ulutabanca et al. [33] described the hormonal deficit in one patient during the acute phase after brain injury. Two studies out of three involving hypogonadism found no linkage between the type of trauma and the presence of central hypogonadism [30,33]. 

Precocious puberty had the lowest prevalence ranging from 1% to 6.5% (rather similar to the general population) [46]. Reviewed studies showed that precocious puberty occurs late after traumatic brain injury (from 6 months to 6 years). Two studies concluded that precocious puberty emerges after severe brain trauma, while the other two mentioned that this complication does not appear to be associated with a certain severity of brain lesions [34,35,36,47]. 

Hyperprolactinemia was reported more frequent in males than in females, most cases in association with hypogonadism [30,31,33,34,35,43]. Even so, these studies could not establish a statistically significant relationship between posttraumatic gonadotropin deficiency and hyperprolactinemia [48]. Hyperprolactinemia was the most frequent finding, with a prevalence of 3.4–64.3%, occurring late after TBI (3 months to 1 year), although Krahulik et al. [34] mentioned two cases (3.4%) with elevated prolactin levels in the acute phase (less than 3 months after trauma), both transient [30,31,33,34,43]. Auble et al. found a causal relationship between moderate to severe brain trauma and elevated prolactin levels, classifying hyperprolactinemia as a persistent complication [31], while other three studies denied a connection between trauma severity and hormonal anomalies, describing hyperprolactinemia as transient [30,33,34]. In our case, the levels of prolactin were normal. 

Central diabetes insipidus secondary to TBI [49] was described with a prevalence of 9.6% in the study conducted by Kaulfers et al. [35]; it appeared only 1 month after brain trauma, it was transient and did not relate to a particular type of injury [35]. Krahulik et al. [34] observed diabetes insipidus was diagnosed in 12 patients during the acute phase, but three months after brain trauma, only one boy still exhibited persistent electrolyte disturbances suggestive of this type of endocrine dysfunction. This study also included four other patients with the syndrome of inappropriate antidiuretic hormone secretion (SIADH), all of which were transient (Table 2).

## 5. Discussion

Although the first description of hypopituitarism as a consequence of TBI is dated almost a century ago, in 1918, when it was reported in a patient with a fractured base of the skull [50], for more than 80 years, it was considered a rather rare condition. However, in the last two decades, numerous studies demonstrated that TBI-related hypopituitarism is frequently found among survivors of TBI [16], with a pooled prevalence of 27% in a meta-analysis that included 19 studies [12,38,51,52,53]. 

While there are many reports about panhypopituitarism after TBI in adults, the pediatric field lacks consistent information [54]. Aoki M. et al. described the case of a 9-year-old boy who suffered from central diabetes insipidus in addition to post-traumatic hypopituitarism [55], and Sayarifard et al. reported a 26-month-old boy who had a prior head injury at the age of 14 months and developed hypopituitarism with low levels of TSH, ACTH, GH, and prolactin [56].

GH deficiency seems to be the most frequent pituitary hormone deficiency in patients tested at least 6 months after the event [16]. In pre-pubertal and pubertal individuals, the isolated presence of GH deficiency can be the cause of permanent short stature with severe consequences in the absence of prompt diagnosis and growth hormone administration due to the effect of sex hormones on the growth plate, which will determine the epiphyseal closure [12,38,51,52,53,56].

Endocrine screening after head trauma is essential but often overlooked [57,58]. This was the case for the teenager we presented. Four years after the severe TBI, he was admitted to our Department for unrelated pathology. The red flags in the case presented were low stature, clinical signs of hypothyroidism and the absence of secondary sex characteristics at the age of almost 18 years. Fortunately, in our patient, due to the presence of both hypogonadism and GH deficiency, the growth prognosis was not severely affected, and the near-final height was close to the average population’s height.

Pediatric prevalence of post-traumatic GH deficiency ranges from 0 to 82% [30,42,56,59], probably due to heterogeneity of the testing methods [34,60]. Thus, in most studies, only patients with decreased levels of IGF-1 and insulin-like growth factor binding protein 3 (IGFBP-3) were subsequently tested for GH deficiency with dynamic tests [31,35,47,59]. Since IGF-1 and IGFBP-3 are not reliable screening tools, some of the patients with GH deficiency were probably missed in these studies [59,61,62]. Moreover, in some of these studies, only non-pharmacological tests were used, like spontaneous nocturnal secretion of GH and/or the cut-offs for GH deficiency were mostly variable and arbitrary [63]. Steroid priming before GH testing in peri-pubertal patients was inconstantly used among studies, introducing another confounding factor in the analysis [64]. Thus, we can conclude that the real incidence of GH deficiency in young patients following TBI is yet to be determined, while further longitudinal studies are required to sustain the long-term surveillance protocols [65].

Post-traumatic hypogonadotropic hypogonadism was reported with a significantly lower prevalence of only 0–12.5% [17], but the available studies are limited by the small number of patients and the lack of prospective data [66]. 

Pediatric secondary hypothyroidism following TBI was reported in a wide range of prevalence (0–64%) [17]. However, this complication might be reversible, as suggested by the study with the greatest number of patients [35]. Thus, in this study, 12/18 males (67%) and 5/13 females (38.4%) were found to have central hypothyroidism [35], but the majority of cases (all but two) were resolved before 12 months [67]. 

Secondary chronic adrenal insufficiency in the post-acute phase of pediatric TBI was found in 0–43.5% [17], but not all the studies used a complete evaluation protocol; thus, the real prevalence might be underestimated. In the study of Dassa et al. [36], a patient with reversible secondary ACTH deficiency after 5.7 years from the injury was reported. The possibility of remission of this type of endocrine anomaly is also supported by the studies in adults, which reported a decreasing prevalence over time, probably due to a certain recovery potential of pituitary cells [42,43]. 

GH and gonadotropin deficiencies are the pituitary syndromes with the most striking clinical manifestation in children. An acute adrenal insufficiency (of a prior chronic mild insufficiency that is not clinically manifested) may be triggered by a certain event like an infection, trauma, or surgery and it might become clinically manifested. That is why these hormonal dysfunctions have the highest probability of being diagnosed; the severity of the previous TBI and the other neurologic, bone or eye sequelae might deviate the attention from the endocrine issues [68]. 

A recent review analyzed 15 pediatric studies, which included a total of 765 patients [17]. These studies showed a predominance of male patients and highlighted the heterogeneity of follow-up testing in relation to the time of injury. In three of these studies, the patients were tested in the acute phase after TBI, while the remainder reported follow-up as far as 10 years post-injury. Taking into account the transience of some post-traumatic hypothalamic-pituitary disorders, we can assume that the exact prevalence of neuroendocrine disorders might be affected by the timing of testing [69,70]. Moreover, the diverse diagnostic criteria used for endocrine entities might further contribute to variate efficacy in the identification of these disorders and sustain practical protocols [23]. It is of massive importance to recognize even the most elusive signs and symptoms of hypopituitarism, especially in a patient with a history of brain injury, because no diagnosis means no treatment, which leads to high morbidity and mortality [71]. That is why screening methods for pituitary dysfunction should be applied to patients who have suffered brain damage, not necessarily for those with mild forms of TBIs (unless they are admitted for suggestive symptomology), but for moderate to severe cases [16]. In 2018, Quinn et al. [16] proposed an algorithm for screening and management of post-traumatic hypopituitarism in cases of moderate or severe TBI. They proposed to check the early morning serum cortisol in the acute phase (days 1–7 post-TBI), then to assess the adrenal, gonadal, and thyroid axes in the chronic phase (3–6 months after TBI) and, additionally, to assess growth hormone axis +/− repeat anterior pituitary assessment at 1 year post-TBI [16]. 

Taking into account the lack of validated data regarding the most appropriate screening strategy for post-acute TBI-related hypopituitarism, the increased awareness of any clinician in any medical field regarding possible panhypopituitarism after head trauma is crucial if the patient is to safely overcome any acute condition considering the severe cortisol deficiency. In our case, orthopedic surgery was only postponed, and a pediatric consult was requested due to increased values of liver enzymes. Acute surgical stress with no glucocorticoid supplementation in a non-diagnosed patient could have been fatal for this teenager [46,72,73]. 

Regarding the prognostic factors for TBI-related hypopituitarism occurrence, some putative risk factors have been suggested [8,74]. Thus, some authors found that severe TBI patients have the highest risk of TBI-related hypopituitarism (35.5%) in comparison with patients with mild (16.8%) and moderate (10.9%) TBI [51] while others found no relationship between severity of TBI and the risk of developing pituitary insufficiency after TBI [11,30,43,75]. It was also suggested that findings at the computed tomography scan, like diffuse brain swelling [76], basal skull fractures, and diffuse axonal injury [77], might be associated with the development of further hypopituitarism, although not all the authors confirmed these results [30,78]. Genetic susceptibility may also be related to the risk of developing hypopituitarism after brain damage, as suggested by a study that finds an association between apolipoprotein E3/E3 genotype and lower TBI-related hypopituitarism incidence [79]. Late-onset anomalies of pituitary hormones require a differential diagnosis from other causes of hypophyseal insufficiency like autoimmune, genetic, etc.

The mechanism of injury may also be associated with TBI–associated hypopituitarism. Blast TBI has been found to confer a higher risk of TBI-related hypopituitarism, based on studies on military personnel returning from war zones [48,80]. Baxter et al. [80] identified that patients who suffered moderate-severe blast TBI had a higher risk of developing hypopituitarism (32%) compared to those with non-blast TBI (2.6%). Undurti et al. [48] found a high prevalence (31%) of TBI-related hypopituitarism among army veterans with prior blast mild TBI, which was associated with increased post-concussive symptoms. 

## 6. Conclusions

In children with previous TBI, pituitary hormone disturbances are common not only during the acute phase but also during the chronic post-traumatic phase. Their occurrence could be clinically evident immediately after trauma or several months, but also years after the brain damage. Although in some patients, the pituitary function can recover, in most patients, the lesions are permanent. Thus, long-term follow-up is important in order to provide adequate care for endocrine complications, which could be life-threatening and impact patients’ overall quality of life. The presented case draws attention to the importance of stature and puberty assessment in pediatric patients with major head trauma, the importance of its recognition in cases that are referred to different types of surgery when it comes to central adrenal insufficiency, the importance of identifying GH deficiency in order to achieve a final height according to genetic and population targets as well as the adequate testosterone replacement in order to help peak bone mass through teens and young adult years.

## Figures and Tables

**Figure 1 diagnostics-13-00212-f001:**
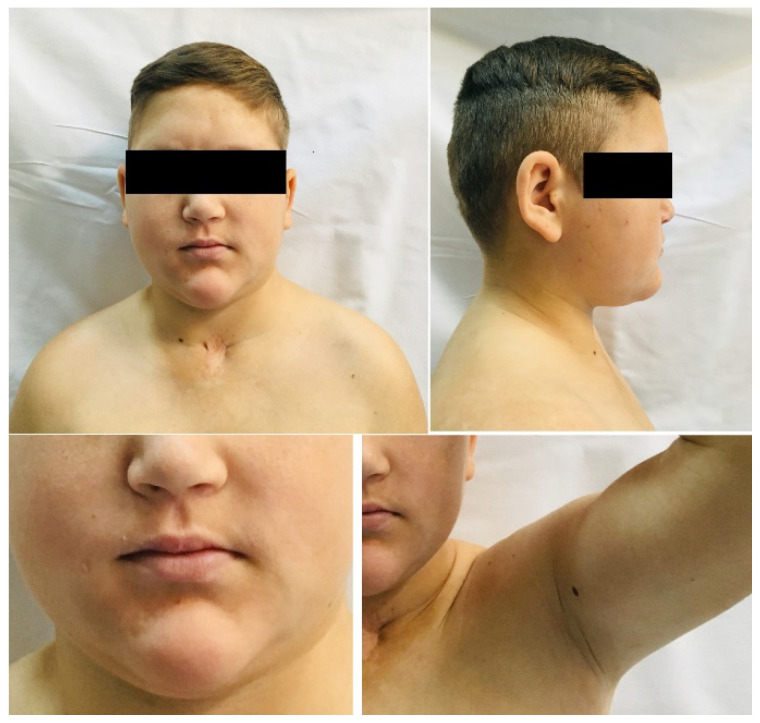
This is a 17-year-old male patient admitted for evaluation of TBI 4 years after a severe car accident. Clinical findings: coarse features, round face, absence of face and axillary hair, delayed puberty.

**Figure 2 diagnostics-13-00212-f002:**
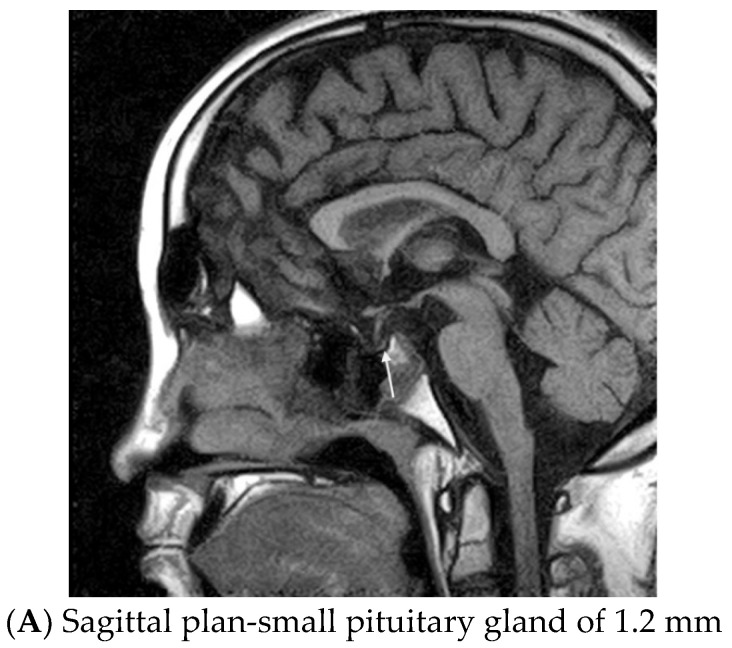
Magnetic resonance imaging on a 17-year-old male with post-brain trauma injury, 4 years after a severe car accident.

**Table 1 diagnostics-13-00212-t001:** A 17-year-old boy: initial hormonal work-up suggestive of panhypopituitarism (4 years after a severe car accident).

Hormone	Value	Normal Ranges
fT4	8.04 pg/mL	9.8–16.3 pg/mL
TSH	0.44 mUI/mL	0.5–4.5 mUI/mL
Serum morning cortisol (8:00 am)	11.91 ng/mL	54–156 ng/mL
ACTH	2.62 pg/mL	7.2–63.3 pg/ml
Serum total testosterone	<0.05 ng/dL	0.3–11 ng/dL
FSH	1.11 mUI/mL	1.5–12 mUI/mL
LH	0.03 mUI/mL	1.3–9.8 mUI/mL
IGF-1	18.9 ng/dL	193–731 ng/dL
prolactin	11 ng/mL	3–26 ng/mL

Abbreviations: fT4: free thyroxine; TSH: thyroid-stimulating hormone; IGF: 1-insulin-like growth factor-1; ACTH: adrenocorticotropic hormone; FSH: follicle-stimulating hormone; LH: luteinizing hormone.

**Table 2 diagnostics-13-00212-t002:** Results of reviewed studies regarding TBI-related hypopituitarism among pediatric patients (please see references [30,31,33,34,35,36,43]).

Authors	Study Type	No. Patients	Median age	Gender Distribution	Trauma Type	Trauma Severity	Time Incident-Diagnosis	Isolated Hormonal Deficit (%)	Multiple Hormonal Deficit (%)	Deficit Type	Prevalence (%)
Agha A. et al. (2004) [30]	Prospective	102		85 M* 17 F*	Road accident Fall Aggression	M** S**		22.5	5.9	Adrenocorticotropic hormone (ACTH) deficiency Gonadotropin (Gn) deficiency Hyperprolactinemia Growth hormone (GH) deficiency	12.7 11.8 11.8 10.7
Auble B.A. et al. (2014) [31]	Retrospective, longitudinal	14	3.1	11 M 3 F	Shaken baby syndrome	M S		86	50	Hyperprolactinemia Thyroid-stimulating hormone (TSH) deficiency Short stature GH deficiency	64.3 42.8 28.6 14.3
Ulutabanca H. et al. (2014) [33]	Prospective	41	7	27 M 14 F	Fall from height Road accident Aggression	M M S	First 24 h to 1 year			ACTH deficiency TSH deficiency Hyperprolactinemia GH deficiency Hypogonadism	26.8 17.1 7.3 4.9 2.4
Krahulik D. et al. (2017) [34]	Retrospective	58	11.3	37 M 21 F	Road accident Tumble Aggression Hanging	M S	First 24 h to 1 year			GH deficiency TSH deficiency ACTH deficiency Hypogonadism Hyperprolactinemia Central diabetes insipidus PP	5.2 5.2 3.4 3.4 3.4 3.4 1.2
Kaulfers A.M. et al. (2010) [35]	Prospective	31	11.6		Road accident Fall Hit by car	M S	1 year			TSH deficiency Hyperprolactinemia Central diabetes insipidus GH deficiency Precocious puberty (PP)	35.4 12.9 9.6 9.6 6.4
Dassa Y. et al. (2019) [36]	Prospective	61		45 M 16 F	Road accident Fall	S	1 year			GH deficiency PP TSH deficiency ACTH deficiency	27.8 6.5 3.2 1.6
Heather N.L. et al. (2012) [47]	Retrospective, longitudinal	198	8.3	112 M 86 F	Accident Aggression Shaken baby syndrome	M S				PP	1

Abbreviations M*-male, F*-female, M-mild trauma, M**-moderate trauma, S**-severe trauma.

## Data Availability

Not applicable.

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
