# Peer review of "An Approach to Traumatic Brain Injury-Related Hypopituitarism: Overcoming the Pediatric Challenges"

_diagnostics, 2023, doi:10.3390/diagnostics13020212_

Round 1

Reviewer 1 Report

The paper is devoted to the important issue of hypopituitarism after traumatic brain injury and in this aspect it should be considered for publication. However, the manuscript in its present form contains some inaccuracies that should be corrected (e.g. the Authors have reported 2-years follow up of the patient – from the age of 17 to 20.5 years).

Case report is clear, however the Table 1 exactly repeats information provided in the main text (except for different IGF-1 values) but both with missing prolactin levels.

Review of the literature and Discussion are extensive, however the key to dividing the information between these two sections is unclear. I would like to suggest the Authors to provide the literature review without addressing to their case as “Review of the literature” and to discuss only the papers related to their case (children, delayed diagnosis of hormonal disturbances) in “Discussion”.

There are also multiple corrections of the text necessary, starting from Affiliations and ending with References. Particularly, the primary source of data concerning the incidence of traumatic brain injury should be quoted. Recording citations in the text and in References should be corrected and unified.

All detailed remarks concerning minor corrections are placed directly as comments on pdf of the manuscript.

So, in my opinion, the paper requires major revision before publication.

Kind regards,

Reviewer

Author Response

Response to Review 1 Comments

Dear Reviewer,

Thank you very much for your time and your effort to review our manuscript.

We are very grateful for providing your valuable feedback on the article.

Here is a point-by-point response and related amendments that have been made in the manuscript according to your review (marked in green color).

  1. Point no. 1

The paper is devoted to the important issue of hypopituitarism after traumatic brain injury and in this aspect it should be considered for publication. However, the manuscript in its present form contains some inaccuracies that should be corrected (e.g. the Authors have reported 2-years follow up of the patient – from the age of 17 to 20.5 years).

Thank you very much. We corrected the period of follow-up.

  1. Point no. 2

Case report is clear, however the Table 1 exactly repeats information provided in the main text (except for different IGF-1 values) but both with missing prolactin levels.

Thank you very much. We reduced the information within the main text; we corrected IGF-1 value, and introduced the prolactin levels (Table 1).

  1. Point no. 3

Review of the literature and Discussion are extensive, however the key to dividing the information between these two sections is unclear. I would like to suggest the Authors to provide the literature review without addressing to their case as “Review of the literature” and to discuss only the papers related to their case (children, delayed diagnosis of hormonal disturbances) in “Discussion”.

Thank you very much. We respectfully choose to overview data from literature in addition to case – related information since this was the aim of the article from the beginning. Moreover, we respectfully introduced data on a larger scale due to the fact that a child with traumatic brain injury might be identified with hypopituitarism as teenager or during transition period or even as young adult thus these aspects should not be restrained to exclusively pediatric population. Even in our case, the actual recognition of hypopituitarism was done during puberty period and he was followed until present day, but in the meantime he became a patient to an adult unit of endocrinology after initial pediatric surveillance. Moreover, there are many endocrinology centers were endocrinologists follow both pediatric and adults patients, too. Thank you.

  1. Point no. 4

There are also multiple corrections of the text necessary, starting from Affiliations and ending with References. Particularly, the primary source of data concerning the incidence of traumatic brain injury should be quoted. Recording citations in the text and in References should be corrected and unified. All detailed remarks concerning minor corrections are placed directly as comments on pdf of the manuscript. So, in my opinion, the paper requires major revision before publication.

Thank you very much. We address each observation as following from Point no. 5 to Point no. 62.

  1. Point no. 5

Please explain the meaning of the stars.

Thank you very much. The stars mean corresponding authors.

  1. Point no. 6

You should give each affiliation separately and two numbers for one author, where necessary.

Thank you very much. We respectfully introduce the affiliations like this (due to the University’ requirements). Moreover, a Department might have the same name if the associated hospital name is not provided (meaning they are not the same).   

  1. Point no. 7

Please use quotation marks in English notation where applicable, e.g. "text"

Thank you. We introduced them.

  1. Point no. 8

Please unify alignment with other affiliations

Thank you. We unified it.

  1. Point no. 9

Should this affiliation consist of the name of institution only?

Thank you. Indeed. We confirm.

  1. Point no. 10

Please start from new page if it is not possible to place any sentence here.

Thank you. We started from a new page.

  1. Point no. 11

This information is cited in Reference [1] but originally it was published in other paper. That paper should be quoted here.

Thank you. We corrected it.

  1. Point no. 12

Please unify spelling "TBI-related" - with or without spaces in the text

Thank you. We choose to use it with space.

  1. Point no. 13

See line 70

Thank you. We corrected it.

  1. Point no. 14

is associated with

Thank you. We corrected it.

  1. Point no. 15

including (what) with late occurrence

Thank you. We corrected it.

  1. Point no. 16

It would be better to introduce abbreviations giving first full names of hormone, followed by an abbreviation in bracket.

Thank you. We switched it.

  1. Point no. 17

Please decide if you list all the numbers of References or use hyphens which are appropriate for 3 and more.

Thank you. We chose hyphens.

  1. Point no. 18

see line 106

Thank you. We corrected it.

  1. Point no. 19

Please explain what tests showed such results.

Thank you. We introduced them.

  1. Point no. 20

Repeated explanation should be remover

Thank you. We removed it.

  1. Point no. 21

Please unify the abbreviation

Thank you. We unified it.

  1. Point no. 22

Missing space in the test and repeated "since"???

Thank you. We corrected it.

  1. Point no. 23

Too much spaces in the text?

Thank you. We removed them.

  1. Point no. 24

Table 1 should be removed if the authors provide the same data in the text or converselu (detailed data removed from the main text and Table 1 left). Please explain the difference in IGF-1 level and provided normal rages in the main text and in Table 1. Please try to put all the Table on one page

We respectfully choose to use the Table, but reduced the information within main text and corrected the values. The Table represents a practical tool of understanding the case. Thank you.

  1. Point no. 25

Range

Thank you. We corrected it.

  1. Point no. 26

abbreviation NR should be introduced if you want to use it

Thank you. We introduced it.

  1. Point no. 27

respectively

Thank you. We corrected it.

  1. Point no. 28

unnecessary spaces? please check all the text for this problem, the same is e.g. in lines 196 and 197

Thank you. We corrected them. However, we respectfully consider these aspects as part of final editing.

  1. Point no. 29

this seems to be not an appropriate term here

Thank you. We corrected it.

  1. Point no. 30

repeated abbreviation not introduced before

Thank you. We corrected it.

  1. Point no. 31

What do the authors exactly mean here? Central nervous system?

Thank you. Indeed, we confirm.

  1. Point no. 32

para-sphenoidal

Thank you. We corrected it.

  1. Point no. 33

remove "aspect" or replace by "picture"

Thank you. We removed it.

  1. Point no. 34

GH values during  ... test were below

Thank you. We corrected it.

  1. Point no. 35

it is not appropriate interpretation as the value of 7 ng/mL refers to maximal GH peak in any of the time.

Thank you. We corrected it.

  1. Point no. 36

follow-up

Thank you. We corrected it.

  1. Point no. 37

to

Thank you. We corrected it.

  1. Point no. 38

All papers concerning ... published ...

Thank you. We changed it.

  1. Point no. 39

please provide missing Reference

Thank you. We introduced it.

  1. Point no. 40

please unify the spelling of abbreviation

Thank you. We unified it.

  1. Point no. 41

please provide the numbers of references directly after the names of authors where appropriate

The statement has only one single reference. This format is accepted, as well. Thank you.

  1. Point no. 42

abbreviation has not been not introduced

Thank you. We corrected it.

  1. Point no. 43

What exactly this Reference refers to? Was the same case report previously published?

No, our case was never published. The similarity of hypopituitarism scenario is mentioned. Thank you

  1. Point no. 44

Why only 2 studies are cited when the authors wrote about 7 studies in this sentence?

Thank you. We corrected it.

  1. Point no. 45

What other anomalies except for TSH deficiency were reported?

Thank you. We already mentioned them, for instance:

“In the study of Auble et al. 17% of the 14 children with moderate-to-severe TBI presented with low peak GH during stimulation tests [31].”

“They found a prevalence of 5% for 256 the GH deficiency at 1 year of follow up, with an increasing trend from 1 month to 6 257 months, but decreasing until 1 year [35].”

“They found that 17 from 61 patients had GH deficiency 1 year after TBI (5 patients had persistent GH 261 insufficiency and a new patient was diagnosed after 6.5 years after TBI) [36].”

  1. Point no. 46

The authors have reported that the patient was diagnosed at the age of 17 years and followed up to the age of 20.5. Is it really 2 years only?

Thank you. We corrected it.

  1. Point no. 47

suffered from

Thank you. We corrected it.

  1. Point no. 48

ACTH surge but not ACTH deficiency may be regarded as a response to stress

Thank you. We respectfully mention that the lack of ACTH surge may be related to stress and/or corticotherapy exposure amid trauma.

  1. Point no. 49

What exactly this "who decreased" refers to?

Thank you. We respectfully mention that we wrote “deceased”.

  1. Point no. 50

it affects ...

We replaced it. Thanks.

  1. Point no. 51

it is unclear and suggests that all References are related to the papers of Krahulik et al., while probably this refers to the prevalence of hyperprolactinemia?

No, we provided different values concerning hyperprolactinemia thus these numbers cannot be provided by the same study, neither this is the case here. Thanks.

  1. Point no. 52

I'm afraid that this information is missing in the description of the case

We introduced the prolactin levels. Thanks

  1. Point no. 53

Do the authors of cited studies differentiate hypogonadism related to hypothalamus or pituitary damage?

No.

Please explain why only hypogonadism was found as caused by hypothalamic injury?

This is not the case. This aspect is part of the section dedicated to hypogonadism. Please see the data we prior mentioned concerning this citation, for instance:

“ACTH deficiency has a prevalence of 1.6 to 26.8% [30,33,34,36]. 288 Agha et al. and Dassa et al. mentioned it was transient and did not appear to relate with a 289 certain type of brain trauma [30,36].”

“Agha et al. included children (and adults): from the 9 patients with early GH 250 deficiency, 5 had recovered within first 6 months, but other 2 patients developed new GH 251 deficiency. At 12 months, only 5 patients had persistent GH deficiency [30].”

  1. Point no. 54

I suggest to explain abbreviations under the Table, narrow this column and re-organize the widths of other columns; alternatively consider reversing the table on a full page

We respectfully consider this aspect is part of the final editing which will be performed in association with a dedicated team of the journal, not part of our reviewing. Thank you very much.

  1. Point no. 55

It should be taken into account that hypogonadism may be secondary to hyerprolactinemia - was this issue discussed in cited papers?

We already mentioned that a clear relationship between hyperprolactinemia and hypogonadism could not be established. Thank you

  1. Point no. 56

Really? The authors have previously reported other problem related to the same accident as a cause of admission?

As we mentioned, the patient was detected with hypopituitarism starting from his admission for correction surgery which was needed after the trauma.

  1. Point no. 57

This was in the case of...

Thanks. We introduced “in”.

  1. Point no. 58

Is this all surneme or surname and name?

This is a good point. Thank you. We removed the surname.

  1. Point no. 59

please correct

Thank you. We corrected it.

  1. Point no. 60

please correct grammatical errors in this very long sentence as it is difficult to understand it; consider to create 2 separate sentences

Thank you. We created two separate senteces.

  1. Point no. 61

not only

Thank you. We introduced it.

  1. Point no. 62

References should be prepared according to the Instruction for Authors - please check it carefully

We adjusted the references that we respectfully consider part of editing process. Thanks

Thank you very much once again for reviewing our paper and for you excellent observations and comments.

Best regards,

The authors

Reviewer 2 Report

Congratulations to the authors for the work. This manuscript is topic of interest for neurosurgeons. It is commendable to present the problem of hypopituitarism following TBI in pediatric

Author Response

Response to Review 2 Comments

Dear Reviewer,

Thank you very much for your time and your effort to review our manuscript.

We are very grateful for your valuable feedback on the article.

Congratulations to the authors for the work. This manuscript is topic of interest for neurosurgeons. It is commendable to present the problem of hypopituitarism following TBI in pediatric

Thank you very much. We appreciate it.

Round 2

Reviewer 1 Report

Thank you for all the corrections done after the first round of review. Unfortunately, some problems have not been resolved, despite the fact that they have been directly indicated (see the list below). Please check all these sentences carefully once more as in its prestent form the manuscript contains some errors that make it still unacceptable for publication.

The authors should unify quotation marks in affiliations according to English punctuation, it should be as in line 23 (however in this line a space is missing after coma). Moreover, affiliations are prepared incorrectly as each author has one affiliation including different institutions, instead of appropriate list of numbers corresponding to the names of particular institutions. Unfortunately the authors have not corrected the affiliations after first review.

The same refers to quotation marks in the main text.

Listing the numbers of references in the main text has not been unified, e.g. [7,8,9] in line 91 and [11,12,13] in line 97 but [17-21] in line 114 and [23-25] in line 118. Unfortunately the authors have not corrected this after first review.

It seems that the horizontal line between lines 154 and 155 should be removed.

The abbreviation of mililiter should be unified in the main text and tables as the authors use “ml” (e.g. with respect to fT4 in Table 1) or “mL” (e.g. in line 160).

The authors have not unified the abbreviation and still use TBI in the main text, while BTI in the description of Figure 1, in line 193. Unfortunately the authors have not corrected this after first review.

The authors should not to repeat the same data on hormonal tests in the main text (lines 172-176 and in the Table (either detailed values in the main text or the whole Table is unnecessary and should be removed). If the authors decide to leave the table, it should not be divided between two pages.

The authors should unify writing the names of hormones, now Adrenocorticotoropic Hormone and Thyroid Stimulated Hormone (lines 106-107), and Growth Hormone (line 115) is written with capital letters , while luteinizing hormone and follicle stimulating hormone (line 175-176) is written with lowercase letters. This should be unified in all the manuscript, e.g. in lines 182-184.

The sentence in lines 217-220 is partially corrected, however it’s not quite in line with the reviewer’s suggestions.

In Discussion, if the names of authors are given in the main text, the numbers of References should be placed directly after these names, not at the end of the sentence or even at the end of paragraph. Unfortunately the authors have not corrected this after first review.

What happened that the authors decided to omit the information on the studies which had not confirmed the relationship between the type of trauma and hormonal disorders (see line 270-271 and previous version of the same sentence)? This is an important information and should not be removed but the References should be refilled.

Line 365 – it should be “in the case of” not “for”. Unfortunately the authors have not corrected this appropriately after first review.

The sentence in line 399-400 has to be corrected once more.

Spelling “follow up” (line 408) and “follow-ups” (line 410) should be unified. Unfortunately the authors have not corrected this after first review.

Additional space(s) in line 420 should be removed.

“TBI – related” in line 435 and in line 450 should be written with spaces.

My suggestion was to replace “both” (highlighted) by “not only” in the line 459, not to include “not only” to the next sentence.

I hope the authors are able to deal with these problems and finally to publish this interesting paper.

Author Response

Response to Review 1 Comments

Dear Reviewer,

Thank you very much for your time and your effort to review our manuscript for the second time.

We are very grateful for providing your valuable feedback on the article.

Here is a point-by-point response and related amendments that have been made in the manuscript according to your second review (marked in yellow color).

Thank you for all the corrections done after the first round of review. Unfortunately, some problems have not been resolved, despite the fact that they have been directly indicated (see the list below). Please check all these sentences carefully once more as in its prestent form the manuscript contains some errors that make it still unacceptable for publication.

The authors should unify quotation marks in affiliations according to English punctuation, it should be as in line 23 (however in this line a space is missing after coma). Moreover, affiliations are prepared incorrectly as each author has one affiliation including different institutions, instead of appropriate list of numbers corresponding to the names of particular institutions. Unfortunately the authors have not corrected the affiliations after first review.

We changed the affiliations. We introduced the space. Thank you

The same refers to quotation marks in the main text.

We corrected them. Thank you

Listing the numbers of references in the main text has not been unified, e.g. [7,8,9] in line 91 and [11,12,13] in line 97 but [17-21] in line 114 and [23-25] in line 118. Unfortunately the authors have not corrected this after first review.

We corrected them. Thank you

It seems that the horizontal line between lines 154 and 155 should be removed.

We removed it. Thank you

The abbreviation of mililiter should be unified in the main text and tables as the authors use “ml” (e.g. with respect to fT4 in Table 1) or “mL” (e.g. in line 160).

We corrected it. Thank you

The authors have not unified the abbreviation and still use TBI in the main text, while BTI in the description of Figure 1, in line 193. Unfortunately the authors have not corrected this after first review.

We corrected it. Thank you

The authors should not to repeat the same data on hormonal tests in the main text (lines 172-176 and in the Table (either detailed values in the main text or the whole Table is unnecessary and should be removed). If the authors decide to leave the table, it should not be divided between two pages.

We corrected it. Thank you

The authors should unify writing the names of hormones, now Adrenocorticotoropic Hormone and Thyroid Stimulated Hormone (lines 106-107), and Growth Hormone (line 115) is written with capital letters , while luteinizing hormone and follicle stimulating hormone (line 175-176) is written with lowercase letters. This should be unified in all the manuscript, e.g. in lines 182-184.

We corrected it. Thank you

The sentence in lines 217-220 is partially corrected, however it’s not quite in line with the reviewer’s suggestions.

We corrected it. Thank you

In Discussion, if the names of authors are given in the main text, the numbers of References should be placed directly after these names, not at the end of the sentence or even at the end of paragraph. Unfortunately the authors have not corrected this after first review.

We corrected them. Thank you

What happened that the authors decided to omit the information on the studies which had not confirmed the relationship between the type of trauma and hormonal disorders (see line 270-271 and previous version of the same sentence)? This is an important information and should not be removed but the References should be refilled.

We reviewed the initial format which was not correct. We respectfully consider this format as adequate. Thank you

Line 365 – it should be “in the case of” not “for”. Unfortunately the authors have not corrected this appropriately after first review.

We corrected it. Thank you

The sentence in line 399-400 has to be corrected once more.

We corrected it. Thank you

Spelling “follow up” (line 408) and “follow-ups” (line 410) should be unified. Unfortunately the authors have not corrected this after first review.

We corrected it. Thank you

Additional space(s) in line 420 should be removed.

We corrected it. Thank you

“TBI – related” in line 435 and in line 450 should be written with spaces.

We corrected it. Thank you

My suggestion was to replace “both” (highlighted) by “not only” in the line 459, not to include “not only” to the next sentence.

We corrected it. Thank you

I hope the authors are able to deal with these problems and finally to publish this interesting paper.

Thank you
